# System Mapping of the Production and Value Chain to Explore Beekeeping Potential in Southwest Ethiopia

**DOI:** 10.3390/insects15020106

**Published:** 2024-02-02

**Authors:** Mulubrihan Bayissa, Ludwig Lauwers, Fikadu Mitiku, Dirk C. de Graaf, Wim Verbeke

**Affiliations:** 1Department of Agricultural Economics, Ghent University, Coupure links 653, B-9000 Gent, Belgium; 2Department of Agricultural Economics and Agribusiness Management, College of Agriculture and Veterinary Medicine, Jimma University, Jimma P.O. Box 307, Oromia, Ethiopia; 3Department of Biochemistry and Microbiology, Ghent University, Krijgslaan 281 S2, B-9000 Gent, Belgium

**Keywords:** beekeeping potential, beekeeping systems, system mapping, value chain, Ethiopia

## Abstract

**Simple Summary:**

The demand for apiary products in a global market is increasing, while the beekeeping sector still faces multiple challenges, in particular in developing countries. For example, Ethiopia, despite belonging already to world’s top producers of honey and beeswax and having favorable agroclimatic conditions and biodiversity, still has a huge unexploited beekeeping potential. However, the beekeeping system remains mostly traditional, so its contribution to the local community and the country’s economy remains minimal. This study aims to explore the development potential of beekeeping by combining systems analysis of the current production with a SWOT-PESTEL analysis. The study identified the core challenges that hinder the exploitation of the potential and the existing opportunities. System mapping revealed that the unproductive practice and orientation of smallholder beekeepers towards poor quality honey production mainly for local beverage making affected the use of the potential and its contribution to beekeepers’ income and the country’s economy. In contrast, the focus of private investors and cooperatives on the production of fully and semi-processed honey for the local and global market had a positive impact on the use of the potential. Providing smallholder beekeepers with training and connecting them to modern input suppliers may help to exploit the existing potential.

**Abstract:**

Ethiopia has a high potential for the production of honey and other apiary products due to its ideal agroecology. This potential is, however, not yet well utilized due to weak production and valorization systems. The study analyzed beekeeping systems and their honey value chain to detect the barriers and to explore ways to better exploit the existing potential. Descriptive statistics, a SWOT and PESTEL matrix, and system mapping were utilized for analysis. Ethiopian beekeeping is still dominated by traditional production systems, followed by modern and transitional systems, differing in types of beehives and the average amount of honey yield. The combined SWOT-PESTEL analysis revealed challenges like a limited supply and high cost of modern beehives, shortage of credit, absence of a honey marketing legal framework, pest and predator attacks, absconding, and uncontrolled application of agrochemicals. Opportunities include the globally increasing demand for honey, availability of good investment policy, conducive agroecology, and support from NGOs. The less productive techniques of smallholder beekeepers’ crude honey production for local beverage making affected the good use of the potential and minimized its contribution to the local and national economy. On the contrary, strengthening private investors and cooperatives towards the production of fully and semi-processed honey impacted the utilization of the potential positively.

## 1. Introduction

Honey bees play a tremendous role in the world as a pollinator, in ecosystem balance, and as a source of income for rural livelihoods and small farms [1]. Honey is a natural food produced worldwide mainly for human consumption for its sweetness, potential health benefits, and as a source of energy [2,3]. China is the leading honey producer in the world, followed by Turkey [4]. In 2021, more than 472,000 tons of honey were produced in China and supplied to the world market [5]. However, the beekeeping sector is facing various challenges [6,7].

Beekeeping, also known as apiculture, is an ancient traditional practice in Ethiopia [8]. It is among the major livestock practices contributing to the economy of the country and income of smallholder beekeepers, besides its environmental protection [9]. Beekeeping plays a crucial role as a livelihood diversification system and in generating revenue for beekeepers [10,11]. It is a promising off-farm activity that directly and indirectly contributes to the livelihood of smallholder rural households and those involved in the value chain [12,13]. Because of honeybees’ contribution to the pollination of crops and vegetables, they also contribute indirectly to boosting crop production and horticultural productivity, as well as environmental conservation [14,15,16,17,18].

Ethiopia is among the top producers of honey and beeswax in the world. Ethiopia ranks first in Africa and it still has a huge beekeeping potential. However, the beekeeping system remains predominantly traditional and, hence, its contribution to the local community, national economy, and the country’s international trade stays minimal [14]. Ethiopia is known for its greatly diversified agroclimatic conditions and biodiversity, which caused the presence of abundant honeybee flora and a massive number of honeybee colonies [19,20,21,22,23]. It is estimated that the country has a total of 10 million *Apis mellifera* bee colonies. All Ethiopian honeybees are grouped into a single subspecies called *Apis mellifera simensis* [14]. Ethiopia has the potential of producing up to 500,000 tons of honey and 50,000 tons of beeswax per annum. However, the current production volume is limited to 53,000 to 58,000 tons of honey and 5742 tons of beeswax [14,24,25,26]. Moreover, out of the total 6,986,100 beehives found in the country, 96% are kept in traditional system, 1% in intermediate system and 3% in modern system beehives [27].

In order to better exploit the available potential and to improve the contribution of the sector to the national economy and rural community, efforts have been made by different stakeholders, but with limited success. Governmental and non-governmental organizations have been working on beekeeping as a means of poverty alleviation, employment creation, and environmental protection. They have been conducting beekeeping development activities like promoting modern beekeeping systems via the dissemination of modern system beehives and their accessories, employing development agents, and promoting honey export activities, but failed to succeed [28,29,30,31]. Thus, the potential is still underutilized due to the poor beekeeping system and various barriers affecting the value chain [32]. The sector still faces various bottlenecks like an inefficient input and output marketing system, lack of market integration and missing institutional arrangements, limited institutional support, access to market, and value chain development [33,34,35]. Moreover, inadequate availability of production technologies, limited modern beekeeping knowledge, limited capacity building, and technical assistance remain major challenges [32,36,37,38,39,40,41].

Given the complexity of the beekeeping sector, system mapping of the production and value chain may help to identify key attention points and to provide recommendations on improvements in weaker links where returns are low [42,43,44]. Mapping the different production systems and the flow of the product through a value chain is important for understanding opportunities and constraints across the chain [45,46,47,48].

Therefore, the overall objective of this study is to explore the development potential of beekeeping in Ethiopia through system mapping of beekeeping and the honey value chain.

## 2. Materials and Methods

### 2.1. Description of the Study Area

The study was conducted in the Jimma zone of the Ormoia national regional state and Kaffa zone of southwest Ethiopia peoples’ national regional state, both located in southwest Ethiopia. These locations are among the top zones with high beekeeping potential in the country as they are endowed with massive natural forests and biosphere reserves. Two districts were selected from each zone: Gera and Gomma districts from the Jimma zone and Gimbo and Shishonde districts from the Kaffa zone (Figure 1).

### 2.2. Data Type, Source and Sampling Techniques

Qualitative and quantitative data from primary and secondary data sources were collected from 15 February to the end of May 2022. The primary data were collected with formal surveys from household heads, honey value chain actors, and agricultural experts at zonal and district levels using pre-tested questionnaires, key informant interviews (KIIs), and focus group discussions (FGDs). Multistage sampling techniques were applied to select the sample respondents. Three-stage sampling was used to select the study participants. In the first stage, three kebeles (peasant associations) from each district were selected using simple random sampling. In the second stage, household heads were stratified into honey producers and non-producers. Finally, 385 heads of honey-producing households were selected using simple random sampling with probability according to the number of honey-producing households in each kebele.

### 2.3. Method of Data Collection

A total of 385 pre-tested survey questionnaires were distributed to gather data from beekeepers. Out of the total questionnaires distributed, 336 were completed and returned to the researcher. Twenty key informant interviews (KIIs) were conducted with beekeeping experts at federal, zonal, and district levels and with development agents (DAs) at the kebele level. Furthermore, twelve focus group discussions (FGDs) were conducted with model beekeepers who are experienced in honey production to identify community level problems related to honey production and marketing. Secondary data, like the number of beekeepers and potential kebeles, types and number of beehives, and types of beekeeping practices in each district, were collected from different reports and databases of district and zonal agricultural offices, including both published and unpublished documents.

### 2.4. Methods of Data Analysis

Data analytical methods include descriptive statistics, value chain, and system mapping, as well as a SWOT and PESTEL analysis. The SWOT (Strengths, Weaknesses, Opportunities, and Threats) analysis comprises four elements or factors that can influence performance and competitiveness. Strengths and weaknesses are internal factors, i.e., they characterize an organization, sector or actor, whereas opportunities and threats are external factors, i.e., they are beyond the control of the concerned organization, sector or actor. PESTEL refers to Political, Economic, Social, Technological, Environmental and Legal analysis. As such, it provides a framework that helps to analyze the macro-environmental factors that can positively or negatively influence actors’ activities and strategies [49], i.e., PESTEL factors may emerge in the SWOT analysis as opportunities or threats. Descriptive statistics, like frequency, mean, minimum, maximum, and percentage, were used to describe the socioeconomic and demographic characteristics of the respondents, to quantify volume of honey produced and marketed from different beekeeping systems, as well as to show the average yield of different types of beehives. Statistical tests such as F-tests and ANOVA with Tukey’s post hoc comparison tests were used to compute the mean difference between selected variables, like types of beekeeping systems and their productivity per hive, average volume of honey produced and sold in kg per household, average number of beehives under different beekeeping systems per household, and income sources of households among the districts. The combination SWOT and PESTEL analysis was used to indicate the internal and external challenges and potential in the honey value chain and to point out possible ways for interventions. Finally, honey production system mapping was used to analyze the different beekeeping systems and their linkages and outputs to identify the strong and weak relationships in order to explore possible actions to better exploit the existing beekeeping potential.

## 3. Results

### 3.1. Demographic and Socioeconomic Characteristics of the Beekeeper Sample

In Table 1, we summarize the sex and marital status of the respondents, who are mostly male (91%) and married (90%).

On average, study participants were 41 years old with an average education level of six grades, ranging from zero (Illiterate) to sixteen (bachelor’s degree) (Table 2). The average family size was four adult equivalents and the average beekeeping experience was 8 years. Beekeepers from different districts significantly differed in age, family size, and beekeeping experience. Gomma district was found to have the highest average beekeeping experience and average age of the respondents.

### 3.2. Honey Production Systems and Beekeeping Practices

#### 3.2.1. Traditional Beekeeping System

Traditional beekeeping is the dominant system in the study area. In this system, the hives are usually hung on a tree in the forest after fumigating the hives with the leaves of a tree with a good smell to attract the colony. Beekeepers also place the hives on a tree in their farmland or backyard or place them under the roof of their house. The community in the study area also conserves the community forest and keeps their hives in the forest.

Within the traditional beekeeping system, the construction materials used for the hives are crucial to classify them, according to the information collected during the FGDs. As such, three types of traditional beehives are found in the study area: log hives, bamboo hives, and softwood hives covered with grass (Figure 2). Log hives are made from cylindrical tree trunks that are divided into two equal parts and carefully curved by deepening the inner part of each half of the log and joining them together. Log hives can be closed at one end and opened at the other to allow the bees to move around and honey harvesting or closed at both ends with only a small hole for the movement of bees. With this type of hive, it is very difficult to inspect the colony and check the status of the honey before harvesting. The hives closed with two ends are very distractive compared to other types of traditional hives, as the beekeeper has to open the whole hive during harvesting. As a result, the beekeeper loses his/her colony completely and damages many bees when opening the hive and crushing the combs. Nevertheless, the log hive is more durable than other traditional hives.

The bamboo traditional hives are prepared from strips of bamboo and internally plastered with cow dung or clay soil and covered with grass, straw or dried leaves of the banana plant on the outside. It is very light compared to other traditional hives. The third type of traditional hive is made of softwood strips or a bundle of thin wood plastered with cow dung or clay soil to make it stronger and more durable. The outer part is covered with grass to protect the bees from rain and to protect them from cold or extreme heat. There is no commonly standardized size for all traditional hives, and their productivity depends on their size. All types of traditional hives have the same oval shape but a different number of openings, depending on the interests of the beekeepers. In all types of traditional hives, the bees build and attach the combs to the wall of the hive; there are no frames inside the hive. The result of the survey shows that 30% of the respondents keep bees only with traditional hives and the majority (87%) construct the traditional hives themselves. Few respondents (13%) bought traditional hives from local markets (Table 3).

The traditional beekeeping system is thus generally characterized by the use of hives built from locally available natural materials, which do not require any special knowledge for their construction or to manage the bees. Beekeeping with traditional hives does not require ownership of land or high initial capital for investment. The disadvantages are that it is difficult to control the bees and determine the condition of the honey for harvesting, the hives are exposed to pests and predators because they are kept in the forest, and their productivity is relatively low. The honey harvested is usually low-quality crude honey, which is a mixture of honeycomb and bee parts, and is not used as table honey but for making local drinks called ‘tej’ (alcoholic honey wine) and ‘birz’ (non-alcoholic honey wine). Other disadvantages pertain to the high rate of colony loss, as well as adverse impacts in terms of deforestation, especially when using log hives. Moreover, it is mainly practiced by men in rural areas, as it is risky and difficult for women to climb into the trees to hang the hives and harvest the honey.
Figure 2Different types of traditional hives. (**A**) Bamboo traditional hives plastered with cow dung; (**B**) log hives; (**C**) traditional hive made from softwood internally plastered with cow dung; (**D**) traditional hive covered with grass and plastic, placed in the garden; (**E**) traditional hive covered with a grass, placed on top of a tree in the forest. Source: Photos taken during data collection, 2022.
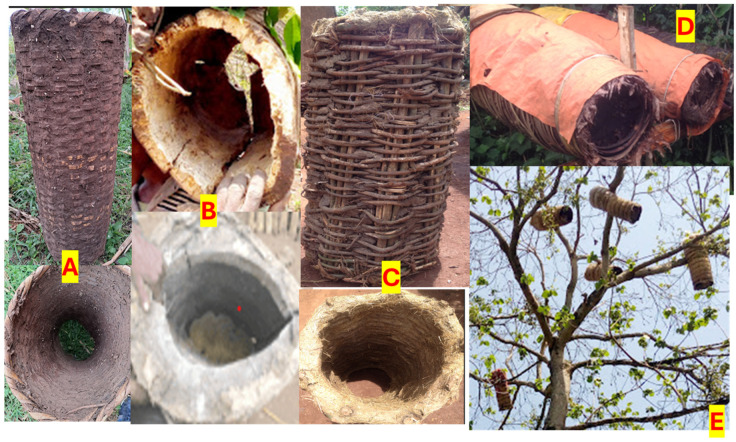


#### 3.2.2. Transitional Beekeeping System

The transitional system of beekeeping is an intermediate system between the traditional and modern systems. The transitional system uses the Kenyan top bar hive and the Chefeka hive, a locally made top bar hive. Beekeepers make the Chefeka hive from wood or bamboo in exactly the same shape and size as the original top bar hive, and plaster the inner part with cow dung and the outer part with mud and cover the upper part with corrugated iron or plastic. Local carpenters also make the top bar hive from lumber and sell it to beekeepers at a relatively low price compared to the box hive. Both types of hives are common in both study areas (Figure 3).

Information obtained from KIIs and FGDs reveals that beekeepers operate with top bar hives around their homestead or backyard for regular inspection. There are also beekeepers who hang their hives in the trees around their houses. The results of the survey show that only a few (3%) beekeepers were using the transitional beehives alone. The majority (69%) of beekeepers’ source top bar beehives from the local market and 24% indicated constructing these hives themselves.

Therefore, the main features of the transitional system of beekeeping are as follows: it is easy to open the hive to inspect the bees and harvest honey, check the status of the honey, and to provide supplementary feed. It is not as destructive as traditional beehives, as the combs can be easily removed at harvest without damaging the colony and disturbing the brood stock. It is relatively productive and produces better quality honey than the traditional system. However, it requires a relatively higher initial capital, as the purchase price is higher, and managing bees with this transitional system requires adequate beekeeping skills.
Figure 3Different types of transitional hives. (**A**) Top bar hive placed in the garden; (**B**) locally prepared top bar hive called Chefeka; (**C**) top bar hive placed under the roof of the house. Source: Photos taken during data collection, 2022.
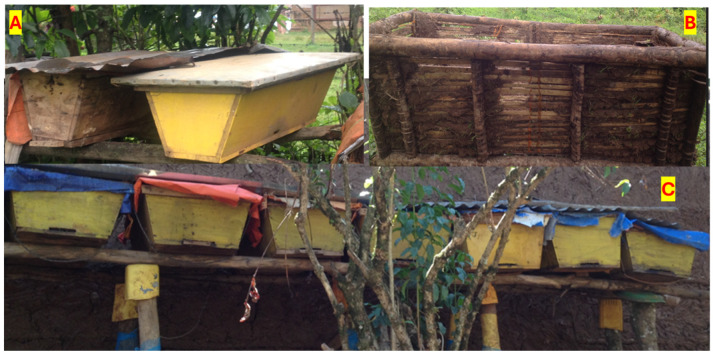


#### 3.2.3. Modern Beekeeping System

The modern beekeeping system is the improved system dominantly used in developed countries and is rarely practiced in developing countries. The modern beekeeping system enables the beekeeper to harvest honey to the maximum degree in line with the colony needs and strength, with high quality, and without harming or stressing the bee colonies too much. The box hive is the most commonly used hive in the modern beekeeping system (Figure 4). The information obtained from FGDs shows that a shortage of box hive supplies exists, and their price render them unaffordable for rural smallholder beekeepers. The majority of beekeepers practicing the modern beekeeping system received the box hives from various supporting organizations for free, on credit or at a subsidized low price. Only a few model beekeepers were able to buy the box hives themselves. The result of the survey shows that 46% of the beekeepers buy the box hives from the local market, 30% from government agencies, and 18% from non-governmental organizations. Moreover, only 13% of the beekeepers used only box hives and 20% used both traditional and box hives.

Beekeepers in the study area mostly used colonies caught from the forest, while it is difficult to catch swarms using box hives. Therefore, beekeepers use traditional hives to capture swarms and relocate these swarms to the box hives after the colony has adapted (information from FGDs). Beekeepers keep the box hives in their garden to closely monitor the colonies and prevent them from being stolen if kept away from the homestead. Hence, the modern beekeeping system is characterized by providing a higher quantity and quality of honey that can be marketed as table honey for the national and international market, being easily manageable, providing enough space for the colony, and ease of inspection and feeding. However, beekeeping with a modern system requires management skills and capital for investment in the hives, which is problematic for rural smallholder beekeepers. Another challenge is that colonies may prefer to stay in traditional hives, which complicates the transfer of swarms after catching them.
Figure 4Modern box hives. (**A**) Box frame hive kept in the garden without a shade; (**B**) box hive placed in the garden under shade. Source: Photos taken during data collection, 2022.
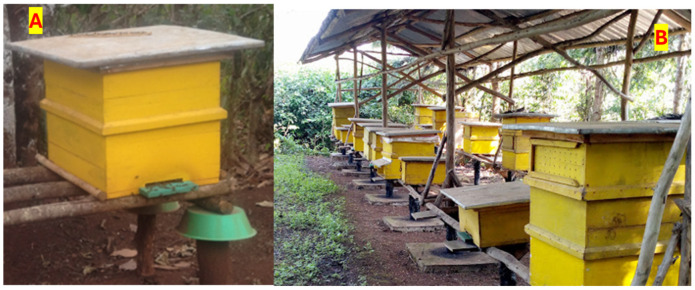


### 3.3. Honey Production and Productivity of Beehives under Different Beekeeping Systems

In the study area, honey and beeswax were the only beekeeping products commercialized as of yet. Almost all beekeepers (96%) were producing only honey, and the remaining 4% were producing both honey and beeswax. During the study period, the average total honey production of traditional, transitional, and modern beekeeping systems was 187 kg, 146 kg, and 183 kg per household per year, respectively (Table 4). Home consumption is important, with a mean volume of 26 kg and a range of 0 to 150 kg per household. The mean volume of honey supplied to the market from traditional, transitional, and modern beekeeping was 170 kg, 144 kg, and 173 kg, respectively. The cumulative average volume of honey produced from the three beekeeping systems per household was 319 kg, while the cumulative average quantity of honey supplied to the market was 292 kg.

The average honey production of the traditional, top bar, and box hives was 11 kg, 20 kg, and 30 kg per hive, respectively. According to insights from the FGDs, the productivity of traditional hives depends on the size of the hive, as there is no standardized size for this type of hive. The size of the log hives is determined by the diameter of the log used by the beekeepers to build it, while the size of the other types of traditional hives is determined by the preferences or habits of the person who built the hive. The F-test statistics for comparing the productivity of the three beekeeping systems show that there is a significant and substantial difference between the districts at the 1% significance level.

On average, households owned twenty-three, four, and five traditional, transitional, and modern hives, respectively. The highest and the lowest number of hives were registered under the traditional and transitional beekeeping systems, respectively. The results of the F-test show that the average total number of hives and the number of hives in each of the three beekeeping systems are significantly different between the districts.
insects-15-00106-t004_Table 4Table 4Beekeeping systems and their productivity, average number of hives owned, and kg of honey harvested and sold from different beekeeping systems per household by district.Types of Beekeeping Systems and Their Productivity in kg Per Hive by District Type of Beekeeping System TotalDistricts 
GeraGommaShishondeGimboMeanMinMaxMeanMeanMeanMeanF-TestTraditional system10.6632512.86 ^a^6.82 ^c^11.97 ^a^9.57 ^b^63.26 ***Transitional system20.0353517.10 ^b^11.06 ^c^24.09 ^a^19.98 ^b^44.59 ***Modern system29.62104528.53 ^b^19.93 ^c^37.80 ^a^31.42 ^b^87.35 ***Average volume of honey produced in kg per household by district Traditional system187.35273.84 ^a^74.69 ^c^163.39 ^b^194 ^b^17.51 ***Transitional system145.8477.95 ^b^63.24 ^b^111.04 a^b^245.93 ^a^4.7 ***Modern system183.48139.31 ^a^184.48 ^a^192.51 ^a^222.96 ^a^0.99Total from all systems318.17325.76 ^a^181.27 ^b^392.36 ^b^385.62 ^b^6.09 ***Honey self-consumed25.5741.14 ^a^16.09 ^b^25.70 ^c^17.69 ^b^23.00 ***Average volume of honey sold in kg per household by districtTraditional system168.67240.42 ^a^66.44 ^c^148.73 ^b^184.74 ^ab^14.88 ***Transitional system144.4169.48 ^b^58.44 ^b^121.04 a^b^233.91 ^a^4.01 ***Modern system173.17118.38 ^a^173.82 ^a^188.52 ^a^211.08 ^a^1.45Total from all systems292.33281.47 ^ab^165.09 ^b^366.83 ^a^367.94 ^a^6.10 ***Average number of beehives under different beekeeping systems per household by district Traditional system hives23.1546.33 ^a^8.08 ^c^17.05 ^b^19.08 ^b^56.15 ***Transitional system hives 3.571.22 ^b^1.12 ^b^3.37 ^b^9.22 ^a^25.83 ***Modern system hives5.103.81 ^bc^6.41 ^ab^7.41 ^a^2.77 ^c^6.62 ***Total number of hives 31.8951.23 ^a^15.79 ^c^28.08 ^b^31.06 ^b^33.67 ***Source: Own survey result from 2022. Note: *** shows the probability level of statistical significance at 1%**.** Mean values with different superscripts, ^a^, ^b^, and ^c^, in the same row are significantly different following ANOVA F-tests and Tukey’s post hoc comparison tests.

### 3.4. Beekeeping as a Household Income Source

Besides honey production, other activities also generate income for the beekeepers in the study area. The beekeepers who participated in the survey primarily cultivate crops like coffee and maize and raise livestock, but also engage in non-agricultural activities such as petty trade, daily outdoor work, and selling firewood or charcoal. Among all income sources, beekeeping is the most important, with a mean annual income of 792.70 USD, while crop and livestock farming are the second and third income sources, with a mean income of 511.98 USD and 165.60 USD, respectively (Table 5). The result of the F-test shows differences in the amounts of total income from livestock, non-agricultural activities, and beekeeping between the districts in the two zones at the 5% significance level.

### 3.5. SWOT and PESTEL Analysis

Data for the SWOT and PESTEL analysis were collected from key informant interviews (KIIs) and focus group discussions (FGDs). The factors affecting honey production and marketing were first categorized into internal (strengths and weaknesses) and external factors (opportunities and threats), and then further divided along their PESTEL area of influence as a second dimension. Both dimensions were structured in a matrix, so the combination of SWOT and PESTEL analysis allowed us to gain extra insights into the internal and external factors influencing honey production and marketing in southwest Ethiopia. These analyses were used to assess the beekeeping business environment and to pinpoint the existing beekeeping potential, current practices, and bottlenecks that can be used as inputs for system mapping to explore the sector for further action. A detailed analysis is provided in Table 6.

### 3.6. Honey Production System Mapping in Southwest Ethiopia

Finally, a system map was drawn to integrate the insights from the previous analyses (Figure 5). As all value chain actors need to be integrated into the overall picture, four levels are distinguished. The focus is on the major honey producers within the three beekeeping systems, namely, the smallholder beekeepers, primary cooperatives, and private investors, and their types of output, namely, crude honey, semi-processed honey, and liquid honey (Figure 1). As mentioned above, beeswax is a product of minor importance, although it has great potential and interest in international markets. Crude honey, a mixture of honey and beeswax, is mainly produced from the traditional beekeeping system and sometimes from the transitional system by smallholder beekeepers. The largest portion of crude honey produced in the study area goes to local beverage production. However, a few rural smallholder beekeepers sometimes semi-process some of the crude honey further using locally available materials to separate honey and beeswax and making the filtered honey available to local consumers. In addition, local retailers semi-process the crude honey they purchased from beekeepers or collectors and sell it to individual households. Fully processed liquid honey comes mainly from modern beekeeping systems and sometimes from transitional systems. This honey is mostly exported to the global market. Private investors and cooperatives are the main sources of liquid honey, as they mainly use modern beekeeping systems and have efficient honey processing facilities.

The systems map shows that crude honey produced through the traditional beekeeping system does not benefit the rural beekeepers in particular and the nation in general to the extent expected. Crude honey is cheaper compared to semi-processed and liquid honey, and it is mainly consumed in the local area in the form of local beverages (tej/birz). Minor benefits are generated from the traditional beekeeping system compared to the efforts made by the smallholder beekeepers and the existing honey production potential
insects-15-00106-t006_Table 6Table 6SWOT and PESTEL matrix of beekeeping, honey production, and apiary product marketing in southwest Ethiopia.
PESTELPoliticalEconomicSocialTechnologicalEnvironmentalLegalSWOT
StrengthsExistence of motivating investment policy Main income source for poor householdsCreate employment opportunities for youth and womenExistence of endogenous beekeeping knowledge Strong motivation for beekeepingAvailability of beekeeping unions and modern processing equipment Environmentally friendly image of beekeepingOrganic nature of the honey in the production areaAbsence of legislation related to the location of hivesWeaknessesAbsence of standardized quality testing labWeak market linkage Excessive cost of the modern hive and its accessoriesShortage of credit High interest ratePoor infrastructure like roads and electricityPoor transportation facilitiesLimited supply of modern beehives and their accessoriesDominance of traditional beekeeping systemsPest and predator attacks and abscondingUncontrolled and unsafe application of herbicides and pesticidesAbsence of a honey marketing legal frameworkIllegal trading during harvestingOpportunitiesTechnical support from the governmentHigh demand from national and international marketsNeeds less initial capital and can be practiced by landless householdsIncreasing trend of honey consumption both domestically and globallyTechnical support from NGOsAvailability of major research centers working on beekeeping technology Conducive agroecology, with abundant bee flora and fauna (e.g., Aphids, which are used to produce honeydew)Existence of apiculture boardPresence of beekeepers’ export associationThreatsPolitical instability High competition from the international marketLocal honey price exceeding the international priceHoney adulteration
Reduction of bee forage resources and bee colonies Incidence of disease and pestsUncontrolled agrochemical application.
Figure 5Honey production system mapping in southwest Ethiopia. Source: Developed by the researchers, 2023.
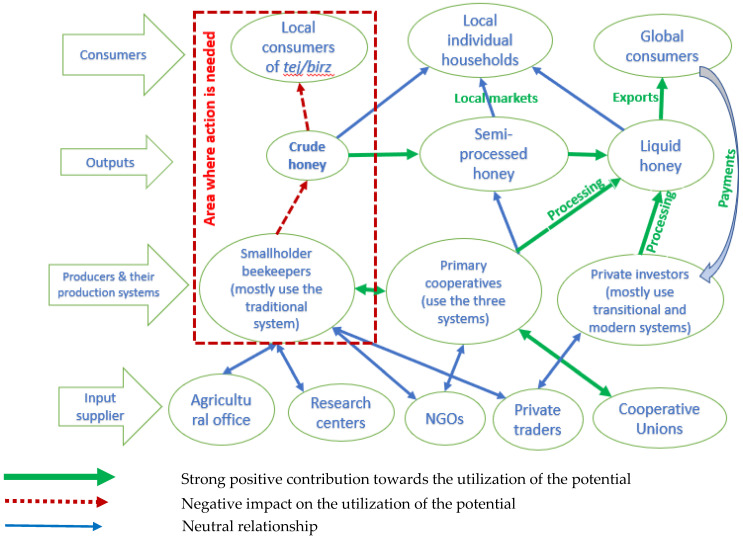


## 4. Discussion

The study found that beekeeping is a significant source of income for rural households in southwest Ethiopia. It contributes to the rural smallholder community’s livelihood because it can be practiced by landless rural households and, besides other farming activities, it does not compete for resources with other agricultural activities like crops and livestock farming, it creates self-employment opportunities for youths and women, and it generates continuous income unlike other seasonal crops. Moreover, it plays a pertinent role in increasing productivity via pollination and in maintaining the ecosystem and poverty reduction. A previous study conducted in Tanzania also provided consistent results [50]. The analysis of the three types of beekeeping systems (traditional, transitional, and modern) observed in the study area corroborates the results of studies conducted in the Amhara region of the Wollo zone [51], in Bako Tibe district of the Oromia regional state [52], in southwestern Ethiopia [53], and in Benishangul-Gumuz, western Ethiopia [54]. Moreover, the review conducted by Teferi [16] reported that traditional (forest and backyard), transitional, and modern systems of honey production are common in rural Ethiopia. The traditional beekeeping system was dominant, followed by the modern and transitional beekeeping systems in the study area. This result is in line with the finding of several previous studies [10,32,36,55].

The study shows a relationship between production systems and the hive type; for example, the choice of the hive material (log, bamboo, or softwood) determines the type of hive used in the traditional beekeeping system. This tight link can be considered as a key factor for further development. In other studies, it was also mentioned that transitional and modern hives, as well as their construction materials, are not easily accessible and affordable compared to traditional hives [14,54,56,57,58]. Moreover, the majority of beekeepers in the study area made the traditional hives themselves from locally available materials, which make them the cheapest and predominantly practiced mainly in rural areas. Nevertheless, their productivity is low compared to top bar and box hives. These low costs combined with acceptable productivity levels and possibly a lack of awareness or belief in the potential of more advanced beekeeping systems make current practices resistant to change. Additionally, as most beekeepers in the study area sourced top bar and box hives from governmental organizations or NGOs in cash or on a credit basis, these more productive beekeeping systems require initial capital to purchase the hives and other inputs, as well as require advanced beekeeping knowledge. Therefore, beekeepers need to be trained in how to manage modern hives. In addition, modern beekeeping is more prone to frequent colony absconding than traditional beekeeping.

Sometimes, there could be a shortage of bee flora around the homestead for the colony, leading to strong competition with other pollinators, which in turn results in lower honey production per colony and also leads to a greater infestation of the bees with pathogens and parasites that can easily spread into the neighborhood. It is, therefore, important to closely monitor colony density, adapt it to the carrying capacity of the natural environment, and to timely provide the bee colonies with supplementary feed if needed, which incurs additional costs. The traditional beekeeping system can, however, be operated with little or no initial capital, which is seen as the main reason why the rural beekeeping sector continues to be dominated by the traditional system, implying underutilization of the beekeeping potential.

Honey is the only commercialized apiary product, and the production of beeswax is still in its early stages in the study area. Other bee products, such as bee pollen, bee venom, royal jelly, and propolis, were not yet known by rural beekeepers and not commercialized in the study area. Earlier studies conducted by Gratzer, Wakjiraet al. [14], Nega [59], and Teferi [16] also reported that, except for honey and beeswax, the other bee products were not yet produced at marketable volume. The average yield of beehives under the three systems in the study area was better than the national average yield and not much lower than the average yield per hive in developed countries. This is a good indicator that southwestern Ethiopia has great potential for honey production, as it is endowed with the natural forests and biosphere reserves mentioned earlier. This finding is in line with the review conducted by Gratzer et al. [14].

The major barriers to honey production in the study area were the high cost and limited supply of modern beehives and their accessories, bee pest and predator attacks, absconding, uncontrolled use of pesticides and herbicides, and a shortage of management skills for modern beehives. This result supports the findings of Benyam, Yaregal et al. [32], Beyene [60], and Ghode [40]. The lack of packing and storage facilities, low honey prices, weak market linkage, weak bargaining power of producers, and shortage of market information are some of the major honey marketing constraints in the study area. The result is consistent with Goshme and Ayele [39] and the study conducted in Kenya by Berem [61]. The main opportunities for honey production were the availability of natural forests and biosphere reserves registered under UNESCO, such as the Kaffa, Shaka, and Yayu biosphere reserves, the presence of abundant bee flora and water, the increasing demand for honey, and the existence of NGOs committed to the development of the beekeeping sector.

Although some findings in this study confirm others from the literature, our more systematic approach of combining SWOT, PESTEL and system mapping allowed for a more proactive analysis of factors that may, or may not, exploit the potential. The four stages of the system analysis in mapping the honey production system have shown that the direct use of crude honey produced by rural smallholder beekeepers under the traditional beekeeping system to produce local beverages (tej/birz) is the main bottleneck. This underutilization of the available potential results in a minimal contribution from the beekeeping sector, because only an insignificant amount passes through the semi-processing stage and reaches the plate of local consumers in the form of table honey. In addition, the system analysis revealed other important ways to exploit the beekeeping potential and improve the contribution of the beekeeping sector to the livelihoods of individual beekeepers and the country’s economy through increasing export volumes and generating foreign exchange. This could be attained through the promotion of primary cooperatives and attracting private investors who use improved beekeeping systems that allow them to produce both fully and semi-processed table honey for local and global markets. Moreover, the ability to process their honey will allow rural beekeepers to benefit from the sale of processed honey and beeswax.

## 5. Conclusions

In order to explore the underutilized beekeeping potential in Ethiopia, the study examined extra information to detect essential challenges for the rural smallholder beekeeping systems in Ethiopia. First, the production structure, dominated by the traditional system, is tied to types of beehives used, productivity differences, and current practices, and is thus rather conservative to changes. Second, the restriction to a few kinds of outputs and valorization chains make the sector less prone to growth.

A combined SWOT and PESTEL analysis and honey production system mapping allowed the identification of anchor points for improvement. Smallholder beekeepers are still largely confined to a low-quality value chain, but there are already several external factors that can help them produce higher-quality products. The most important is increasing the involvement of rural smallholder beekeepers in modern beekeeping systems by facilitating access to modern beekeeping equipment and organizing beekeepers into primary cooperatives and associations for the production of fully processed liquid honey and semi-processed honey. Supporting smallholder beekeepers in the production of at least semi-processed honey is another action point to generate more profit from the sale of honey and beeswax. In addition, the involvement of private investors in the production of fully processed liquid honey that targets consumers at the global level has a positive impact on the utilization of the existing potential and generates foreign exchange to increase the sector’s contribution to the country’s economy. Strengthening the capacity of input suppliers who provide key inputs to beekeepers will also help to better utilize the existing potential.

## Figures and Tables

**Figure 1 insects-15-00106-f001:**
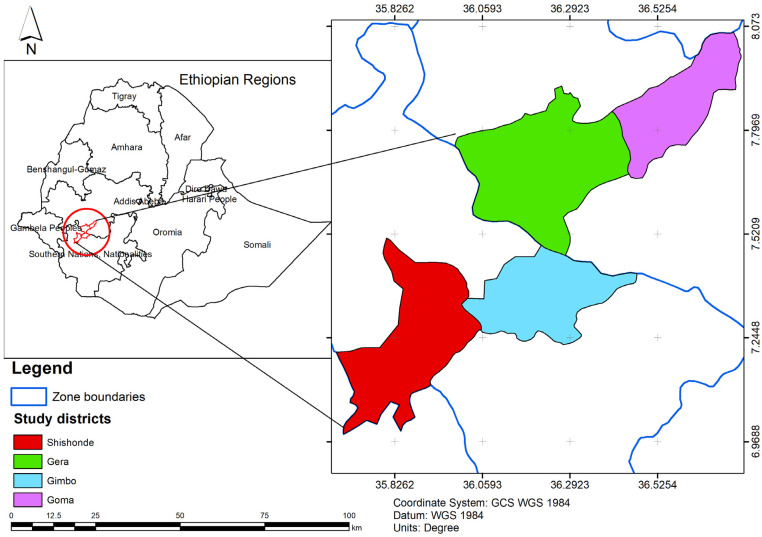
Administrative map of the study area. Source: Own sketch, 2023.

**Table 1 insects-15-00106-t001:** Demographic characteristics of sample households for categorical variables by district.

Variables	Jimma Zone		Kaffa Zone		Total (*n* = 336)
	Gera District (*n* = 91)	Gomma District (*n* = 86)	Shishonde District (*n* = 81)	Gimbo District (*n* = 78)	
	Freq. (%)	Freq. (%)	Freq. (%)	Freq. (%)	Freq. (%)
Sex of the respondents					
Female	5 (5.5)	9 (10.5)	8 (9.9)	7 (9)	29 (8.63)
Male	86 (94.5)	77 (89.5)	73 (90.1)	71 (91)	307 (91.37)
Marital status of the respondent					
Single	4 (4.4)	2 (2.3)	10 (12.3)	8 (10.3)	24 (7.1)
Married	85 (93.4)	81 (94.2)	70 (86.4)	65 (83.3)	301 (89.6)
Widowed	2 (2.2)	1 (1.2)	1 (1.2)	2 (2.6)	6 (1.8)
Divorced	0 (0.0)	2 (2.3)	0 (0.0)	3 (3.8)	5 (1.5)

Source: Own survey results from 2022.

**Table 2 insects-15-00106-t002:** Demographic characteristics of household heads for continuous variables by district.

Variables	Total				Jimma Zone		Kaffa Zone		
					Gera District	Gomma District	Shishonde District	Gimbo District	
	Mean	Min	Max	Std. Dev.	Mean	Mean	Mean	Mean	F-Test
Age	41.08	20	65	8.91	41.22	42.99	41.32	41.32	3.39 **
Education level	6.15	0	16	4.08	5.43	6.10	6.37	6.37	2.00
Family size	3.86	1	12.58	1.59	3.51	2.91	3.25	3.25	2.18 *
Experience	8.19	2	30	4.26	8.12	9.31	8.41	6.82	4.92 ***

Source: Own survey results from 2022. Note: *, ** and *** show the probability level of statistical significance at 10%, 5%, and 1%.

**Table 3 insects-15-00106-t003:** Types and sources of beehives used and location of beekeeping practices.

Types of Beehives Used by Beekeepers		Total		Proportion of Practice by District in Percentage			
		Freq.	%	Gera	Gomma	Shishonde	Gimbo
Traditional only		102	30.40	50	33	5	32
Top bar only (transitional)		9	2.70	1	0	0	10
Box hive only (modern)		43	12.80	7	34	34	3
Traditional & box hive		60	17.90	22	16	16	6
Traditional & top bar		40	11.90	6	1	1	30
Top bar & box hive		10	3.00	2	6	6	0
All three types		72	21.40	13	11	11	19
Sourcing of traditional beehives							
Own construction		237	86.50	29.1	16.5	28.7	25.7
Local market		37	13.50	35.1	35.1	10.8	18.9
Sourcing of transitional beehives							
Own construction		32	23.88	18.8	6.3	21.9	53.1
Local market		93	69.40	16.1	11.8	46.2	25.8
Government office		7	5.22	0	42.9	28.6	28.6
NGO		2	1.49	0	0	0	2
Sourcing of modern beehives							
Own construction		2	1.09	0	0	100	0
Local market		85	46.20	7.1	30.6	49.4	12.9
Cooperatives		8	4.35	0	0	62.5	37.5
Government office		55	29.89	43.6	49.1	3.6	3.6
NGO		34	18.47	26.5	8.8	47.1	17.6
Location of beekeeping practice by district							
Backyard	No	102	30.40	46	27	16	31
	Yes	234	69.60	54	67	84	69
On farmland outside their own backyard	No	284	85.00	93	72	80	92
	Yes	52	15.00	7	28	20	8
In forests	No	107	32.00	14	50	31	33
	Yes	229	68.00	86	50	69	67
On rented land	No	238	98.00	100	99	99	92
	Yes	8	2.00	0	1	1	8

Source: Own survey results from 2022.

**Table 5 insects-15-00106-t005:** Income sources of households in 2021 in USD.

Sources of Income	Overall Total			Jimma Zone		Kaffa Zone		
				Gera	Gomma	Shishonde	Gimbo	
	Min	Max	Mean	Mean	Mean	Mean	Mean	F-Test
Livestock income	0	1182.75	165.60	168.67 ^ab^	103.57 ^b^	178.15 ^ab^	217.39 ^a^	3.19 **
Crop income	0	6935	511.98	589.47 ^a^	475.67 ^a^	379.01 ^a^	599.68 ^a^	1.59
Off-farm income	0	1520	38.61	33.36 ^ab^	76.49 ^a^	18.08 ^b^	24.29 ^ab^	2.94 **
Beekeeping income	22.80	18,939.20	792.70	678.76 ^ab^	576.90 ^b^	780.96 ^ab^	1175.74 ^a^	3.15 **

Source: Own survey result from 2022. Note: ** indicates a probability level of statistical significance at 5%. Mean values with different superscripts, ^a^ and ^b^, are significantly different following ANOVA F-tests and Tukey’s post hoc comparison tests.

## Data Availability

Data will be made available on request.

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
