# Peer review of "System Mapping of the Production and Value Chain to Explore Beekeeping Potential in Southwest Ethiopia"

_insects, 2024, doi:10.3390/insects15020106_

Round 1

Reviewer 1 Report

Comments and Suggestions for Authors

Dear Author, I found your paper very interesting and I don’t have any particular remark to do. Only few small imperfections and one clarification:  

Pg 2 line 51: ..more than 472,000 metric tons of honey were..: more than 472,000 tons of honey (why metric?)

Pg 10 table 6: what do you mean by “bee floras and faunas”? floras is clear, but with faunas do you mean the insects (aphides) which bees exploit to produce honeydew? Or something different? Please explain

Pg 11 Fig. 1: birth: birz

Pg 12 Line 386: Bako tibe: Bako Tibe

PG 15 References

In general, some references are lacking data useful to find them or are written not in accordance with the journal directives, please check carefully. For example:

Line 501 reference 5: the reference lacks the name of the journal and issue: Viktória, V. and F.F. Aliz, Trends in Honey Consumption and Purchasing Habits in Some European Countries, 2023

I found this: …..Applied Studies in Agribusiness and Commerce 17(1)

DOI: 10.19041/APSTRACT/2023/1/6

Author Response

Dear Reviewer,

Thank you very much for taking you time to review this manuscript. Kindly find the detailed responses attached herewith. 

Reviewer 2 Report

Comments and Suggestions for Authors

The manuscript titled “Production and Value Chain System Mapping to Explore the Potential of Beekeeping in Southwest Ethiopia” was reviewed. The manuscript highlighted the strong tradition of beekeeping in Ethiopia and tried to find the potential of maximum profit from beekeeping. However, the problem seems to be more complex and I don't see a higher profit just using modern box hives and honey processing. You state that beekeeping in Ethiopia is an ancient traditional practice. Local people are familiar with beekeeping in their traditional beehives. On the other hand, beekeeping with modern box hives needs much more effort and, above all, specific knowledge. This would primarily mean improving the beekeeping knowledge of the local people, as modern box hives and equipment are very expensive and can be very unprofitable with inappropriate beekeeping practices. And there is always the risk of theft. You also mentioned the potential for honey production of up to 500,000 tons per year. It is necessary to take into account that bee colonies in box hives are much stronger, and therefore consume even more honey and pollen. Keeping the same number of bee colonies and increasing their strength could therefore lead to a local overpopulation of the landscape and a shortage of pollen and nectar sources, which means stress for the honey bees. That will reduce honey production per colony and also leads to a more extensive infestation of bees with pathogens and parasites that can easily spread to the surroundings. In this case, there can be competition between bee colonies and even between other pollinating insect species, which can lead to disruption of biodiversity and stability. From this point of view, it would be appropriate to discuss in the manuscript the need for additional education of beekeepers, as well as the risks of bee diseases, pesticide intoxication, residues in bee products, or complete loss of annual income from beekeeping. It also would be good to supplement the descriptions of the types of hives with their photos.

Author Response

Dear Reviewer,
Thank you very much for taking you time to review this manuscript. Kindly find the detailed responses attached hereunder. 

Reviewer 3 Report

Comments and Suggestions for Authors

Thank you for your manuscript which is really good, I let you some suggestion to improve its quality.

After the reading I am wondering of several things that could discussed or mentionned in your work:

1) What is the colony density in Ethiopia ? Given the different climatic conditions in the country, the results of your study can be transposable to the other regions ? The honey bee has still the same biology ?

2) Now, we know that massively introduce native honey bee colonies may negatively impact other insect pollinators by competition and disease transfer especially if considering modern beekeeping. What is your point of view for this ? Wouldn't it be better to keep traditional, local beekeeping, which seems to be more respectful of the environment? Instead of promoting more professional beekeeping, even if I understand the opportunities it can provide. 

3) In the results part, it could be nice to better synthetize the information.

Here you will find more particular suggestions:

Introduction part:

You are in an insect journal, so it could be interesting to introduce the specific honey bee subspecies which is producing the apiary product towards all the country.

In M&M, a map of the experimental must be added to clearly understand where are your study sites.

118-119: maybe change this : "with probability proportional" ==> "according to"

132: Can you develop more deeply what is the SWOT and PESTEL analysis ?

139: Change "charastics" by "variables". Can you remind also what are these variables ? 

137 : What is the difference between t-test and F-test ? Do you check all the application conditions of all your test (i.e., independance of your replicates, normality of your variables/data....)

Table 2 : What is the scientific interest/motivation to compare the four regions by statistical test ?

161-164 : This needs to go in M&M or to be removed

168-169: Can you add a picture of this in your manuscript ? And for the three

174-175: Can you add pictures for this ?

236: The disadvantages of modern beekeeping are also related to the biology of the subspecies colony ? High swarming frequence ? Aggresivity of the colony ? Can you develop this ? In the dsicussion maybe

Table 4: In which units are expressed the  productivity per hive. Also here why to compare value by F-stat ? What is the number of colonies owned by the smallholders ?

304 : Maybe convert all the birr money to US dollar please.

Author Response

(The authors gave the same response as above.)

Round 2

Reviewer 2 Report

Comments and Suggestions for Authors

Dear authors,

after this revision I recommend the manuscript for publication.

Author Response

Dear Reviewer,

Many thanks for taking the time to review this manuscript. Please find the detailed responses below and the corresponding revisions/corrections highlighted/ in the re-submitted files.